# Simultaneous Inhibition of Ceramide Hydrolysis and Glycosylation Synergizes to Corrupt Mitochondrial Respiration and Signal Caspase Driven Cell Death in Drug-Resistant Acute Myeloid Leukemia

**DOI:** 10.3390/cancers15061883

**Published:** 2023-03-21

**Authors:** Kelsey H. Fisher-Wellman, Miki Kassai, James T. Hagen, P. Darrell Neufer, Mark Kester, Thomas P. Loughran, Charles E. Chalfant, David J. Feith, Su-Fern Tan, Todd E. Fox, Johnson Ung, Gemma Fabrias, Jose’ Luis Abad, Arati Sharma, Upendarrao Golla, David F. Claxton, Jeremy J. P. Shaw, Debajit Bhowmick, Myles C. Cabot

**Affiliations:** 1Department of Integrative Physiology and Metabolism, Brody School of Medicine, East Carolina University, Greenville, NC 27834, USA; 2East Carolina Diabetes and Obesity Institute, East Carolina University, Greenville, NC 27858, USA; 3UNC Lineberger Comprehensive Cancer Center, University of North Carolina at Chapel Hill School of Medicine, Chapel Hill, NC 27599, USA; 4Department of Biochemistry and Molecular Biology, Brody School of Medicine, East Carolina University, Greenville, NC 27834, USA; 5Department of Medicine, Hematology/Oncology, University of Virginia School of Medicine, Charlottesville, VA 22903, USA; 6University of Virginia Cancer Center, Charlottesville, VA 22908, USA; 7Department of Cell Biology, University of Virginia, Charlottesville, VA 22903, USA; 8Research Service, Richmond Veterans Administration Medical Center, Richmond, VA 23298, USA; 9Department of Pharmacology, University of Virginia School of Medicine, Charlottesville, VA 22904, USA; 10Department of Microbiology, Immunology, and Cancer Biology, University of Virginia School of Medicine, Charlottesville, VA 22908, USA; 11Research Unit on Bioactive Molecules (RUBAM), Department of Biological Chemistry, Institute for Advanced Chemistry of Catalonia (IQAC), Spanish Council for Scientific Research (CSIC), Jordi Girona 18-26, 08034 Barcelona, Spain; 12Penn State Cancer Institute, Hershey, PA 17033, USA; 13Department of Pharmacology, Penn State College of Medicine, Hershey, PA 17033, USA; 14Division of Hematology and Oncology, Penn State Cancer Institute, Hershey, PA 17033, USA; 15Department of Experimental Pathology, University of Virginia School of Medicine, Charlottesville, VA 22904, USA; 16Flow Cytometry Division, Brody School of Medicine, East Carolina University, Greenville, NC 27834, USA

**Keywords:** acute myeloid leukemia, sphingolipids, P-glycoprotein, chemotherapy resistance, ceramide

## Abstract

**Simple Summary:**

The sphingolipid ceramide is a key player in cytotoxic responses elicited by many anticancer drugs. Confounding this asset, however, are enzymes that promote ceramide clearance, thus limiting the propagation of ceramide-driven cancer cell death. Because several key ceramide-metabolizing enzymes have been shown to be upregulated in chemotherapy-resistant acute myeloid leukemia (AML) cells compared to chemotherapy-naïve counterparts, the current study was designed to determine the effects of blocking ceramide clearance in drug-resistant AML. For this, we employed simultaneous inhibition of ceramide hydrolysis and ceramide glycosylation and demonstrated that this dual blockade produced multi-fold elevations in intracellular ceramide levels, corrupted mitochondrial function, enhanced caspase activation, and elicited cell death in models of drug-resistant AML. We have herein identified sphingolipid metabolic junctures that can be targeted to enhance leukemia cell vulnerability in the drug-resistant setting, thus providing a novel therapeutic modality in difficult-to-treat cancers.

**Abstract:**

Acute myelogenous leukemia (AML), the most prevalent acute and aggressive leukemia diagnosed in adults, often recurs as a difficult-to-treat, chemotherapy-resistant disease. Because chemotherapy resistance is a major obstacle to successful treatment, novel therapeutic intervention is needed. Upregulated ceramide clearance via accelerated hydrolysis and glycosylation has been shown to be an element in chemotherapy-resistant AML, a problem considering the crucial role ceramide plays in eliciting apoptosis. Herein we employed agents that block ceramide clearance to determine if such a “reset” would be of therapeutic benefit. SACLAC was utilized to limit ceramide hydrolysis, and D-*threo*-1-phenyl-2-decanoylamino-3-morpholino-1-propanol (D-*threo*-PDMP) was used to block the glycosylation route. The SACLAC D-*threo*-PDMP inhibitor combination was synergistically cytotoxic in drug-resistant, P-glycoprotein-expressing (P-gp) AML but not in wt, P-gp-poor cells. Interestingly, P-gp antagonists that can limit ceramide glycosylation via depression of glucosylceramide transit also synergized with SACLAC, suggesting a paradoxical role for P-gp in the implementation of cell death. Mechanistically, cell death was accompanied by a complete drop in ceramide glycosylation, concomitant, striking increases in all molecular species of ceramide, diminished sphingosine 1-phosphate levels, resounding declines in mitochondrial respiratory kinetics, altered Akt, pGSK-3β, and Mcl-1 expression, and caspase activation. Although ceramide was generated in wt cells upon inhibitor exposure, mitochondrial respiration was not corrupted, suggestive of mitochondrial vulnerability in the drug-resistant phenotype, a potential therapeutic avenue. The inhibitor regimen showed efficacy in an in vivo model and in primary AML cells from patients. These results support the implementation of SL enzyme targeting to limit ceramide clearance as a therapeutic strategy in chemotherapy-resistant AML, inclusive of a novel indication for the use of P-gp antagonists.

## 1. Introduction

Although venetoclax is the most important current drug for treating acute myelogenous leukemia (AML), a chemotherapy regimen consisting of daunorubicin (DNR) and cytarabine (Ara-C) is the cornerstone of induction therapy in AML, a hematological malignancy marked by the accumulation of large numbers of immature myeloblasts in bone marrow. In AML, the overall prognosis is poor. The incidence of AML increases with age, and most cases occur in patients > 60 years of age. Unfavorable karyotypes are common with antecedent myelodysplastic syndromes in older patients. In patients receiving therapy with curative intent, less than one half will achieve long-term survival. The advent of targeted therapies for leukemia, such as the first-in-class, orally bioavailable BCL-2 inhibitor, venetoclax, has changed the therapy landscape; however, even these new agents are not without shortcomings [1]. Chemotherapy resistance, which can be intrinsic or acquired, is the major cause of cancer treatment failure, as most patients subsequently relapse with a chemo-resistant form of the disease that is difficult to treat and manage. Given the high percentage of morbidity in AML patients, the development of new therapeutic approaches to improve quality of life and durability of response is critical.

Resistance to drugs with diverse chemical structures and mechanisms of action is known as multidrug resistance (MDR) [2]. In leukemia as well as other neoplasms, resistance mechanisms that involve membrane-resident proteins belonging to the ATP-binding cassette (ABC) transporter protein family, such as P-glycoprotein (P-gp), are impactful [3]. In this scenario, enhanced expression of these proteins is often associated with poor prognosis and frequent relapsed or refractory disease [4]. ABC transporters act as ATP-dependent efflux pumps and reduce the intracellular concentrations of anticancer agents, greatly diminishing clinical efficacy. In addition to drug efflux, other factors contribute to drug resistance, inhibition of apoptosis, DNA damage repair, drug inactivation, and dysfunctional sphingolipid (SL) metabolism with accompanying mitochondrial alterations [5,6]. Moreover, P-gp plays a prominent role in the glycosylation of pro-apoptotic ceramide [7,8], blunting cell death cascades.

Ceramide, a pro-apoptotic tumor-suppressing SL, is the hydrophobic backbone of most SLs and glycosphingolipids [9]. Historically, some of the first evidence of chemotherapy interaction with SL metabolism was reported by Bose et al. [10] and Jaffrezou et al. [11]. Using leukemia cell models, these investigators observed that DNR stimulated ceramide formation and elicited apoptosis via the de novo pathway by activating ceramide synthase and sphingomyelin hydrolysis. With ceramide playing a central role in the induction of apoptosis [12], it is easy to envision that cancer cell defense mechanisms might employ conversion and/or destruction of ceramide for survival. For example, upregulated ceramide glycosylation has been identified as a marker of MDR in leukemia [13], and upregulated acid ceramidase (AC) expression is prominent in AML [14].

In recent works [5,15], we evaluated the possible association of SL metabolism with drug resistance in AML, wherein we showed that chemotherapy selection pressure (DNR, vincristine, VCR) enhanced expression of AC, glucosylceramide synthase (GCS), and sphingosine kinase 1 (SPHK1), enzymes that catalyze ceramide hydrolysis, glycosylation, and formation of sphingosine 1-phosphate (S1P), respectively. These enzyme changes support ceramide elimination and provide mitogenic opportunity via S1P generation. Profound alterations in mitochondrial quality and quantity also accompanied chemotherapy selection pressure, lending further credence to the idea that mitochondrial remodeling hallmarks AML drug resistance.

Seizing upon our observation that increased GCS and AC expression parallels the generation of the MDR phenotype, the goal of the present study was to determine whether targeting these enzyme junctures would provide an effective therapeutic avenue in the drug resistance setting. The intent was to limit ceramide elimination and thus provide an inroad to potentiate ceramide-orchestrated cancer cell death [12]. To that end, dual blockade of AC and GCS in MDR leukemia models resulted in profound increases in cellular ceramide levels that were accompanied by apoptotic cell death involving inactivation of Akt, GSK-3β activation, and increases in the Bcl-2 family member Mcl-1S, the pro-apoptotic splice variant of anti-apoptotic Mcl-1L, and decisive declines in mitochondrial respiratory kinetics. The combination was effective in primary AML patient samples and an HL-60/DNR xenograft model.

## 2. Materials and Methods

### 2.1. Materials

SACLAC (2-chloro-*N*-(2S,3R)-1,3-dihydroxyoctadecan-2-yl acetamide), the AC inhibitor, was obtained from the Research Unit on Bioactive Molecules, Institute for Advanced Chemistry of Catalonia (Barcelona, Spain). The GCS inhibitor D-*threo*-1-phenyl-2-decanoylamino-3-morpholino-1-propanol (D-*threo*-PDMP) was purchased from Matreya and dissolved in DMSO. Eliglustat, a specific GCS inhibitor, was from Selleckchem. Daunorubicin (hydrochloride) and vincristine sulfate were obtained from R&D Systems. The Caspase-Glo^®^ 3/7 Assay kit from Promega was used as directed; assays were conducted in white, 96-well plates (Corning, Thermo Fisher Scientific, 168 Third Avenue, Waltham, Massachusetts, 02451 USA). Rhodamine 123 was purchased from Acros Organics. FITC Annexin V, Annexin V binding buffer, and DAPI were purchased from Biolegend.

### 2.2. Cell Culture, AML Cell Lines, and Establishment of Daunorubicin Resistant HL-60 Cells

Human AML cell lines HL-60, KG-1a, and MV4-11 were obtained from ATCC. MOLM-13 cells were from H-G Wang, Penn State College of Medicine, Hershey, PA, USA. MOLM-14 cells were from Mark Levis, Johns Hopkins University School of Medicine, Baltimore, MD, USA. Wild-type (wt) parental cell lines were authenticated by documentation provided by ATCC, which included antigen expression, DNR profile, short tandem repeat profiling, and cytogenetic analysis. Documentation for other wt cell lines was provided by the contributors. Cells were maintained in RPMI-1640 medium (Life Sciences Technology, Thermo Fisher Scientific, 168 Third Avenue, Waltham, MA 02451 USA) supplemented with 10% fetal bovine serum (FBS) (Peak Serum, Inc., 6598 Buttercup Dr Unit #3, Wellington, CO 80549 USA) and 100 units/mL penicillin and 100 µg/mL streptomycin. HL-60 wt cells were selected for DNR resistance via continuous, long-term exposure to stepwise increments in drug concentrations initiated with 10 ng/mL and ending with 1.0 µg/mL [5]. The resistant cells, designated HL-60/DNR, were maintained in a medium containing 1.0 µg/mL DNR; however, experiments were conducted in a DNR-free medium. The human, VCR-resistant (multidrug-resistant) cell line, HL-60/VCR, was provided by A.R. Safa, Indiana University School of Medicine, Indianapolis, IN, USA. Cells were cultured in RPMI-1640 medium supplemented with 10% FBS, 100 units/mL penicillin, 100 µg/mL streptomycin, and 1.0 µg/mL VCR sulfate.

### 2.3. Cell Viability Assays

Cell viability was determined by fluorescence measurement, as previously described [15]. Cell viability was also measured by trypan blue exclusion. For this procedure, a Countess II automated cell counter was used (Invitrogen, Thermo Fisher Scientific, 168 Third Avenue, Waltham, MA 02451 USA), with disposable hemocytometers, following the manufacturer’s instructions. For determining cellular resistance to DNR and VCR, the CellTiter 96 Assay Kit (MTS), Promega (Promega Corporation, 2800 Woods Hollow Road, Madison, WI 53711 USA), was used according to the manufacturer’s instructions.

### 2.4. AML Patient Samples and Patient Cell Viability Assay

Primary AML cells were acquired from de novo, untreated AML patients. Inclusion and exclusion criteria: de novo, untreated patients, all demographics included (ethnicity, race, gender/sex as a biological variant), age range 35–80 y.o. Peripheral blood mononuclear cells were isolated by Ficoll-Paque (GE Healthcare Life Sciences, Marlboro, MA, USA) density gradient centrifugation. Cells were cultured in RPMI-1640 (Corning) containing 20% FBS (VWR, Avantor, 100 Matsonford Road, Radnor, PA 19087-8660 USA) and incubated at 37 °C and 5% CO_2_. Informed consent was obtained from all patients under the University of Virginia Institutional Review Board and Penn State College of Medicine Institutional Review Board-approved protocol according to the Declaration of Helsinki.

Primary AML cells were plated at 5000 cells/well in tissue culture-treated 384-well plates (Thermo Fisher Scientific) and incubated with DMSO vehicle, SACLAC, D-*threo*-PDMP, or combination at the indicated doses for 48 h. Cell viability was assessed using the CellTiter-Glo luminescence viability assay (Promega), as per the manufacturer’s protocol. Luminescence was measured using the GloMax Discover (Promega). Background media signal was subtracted from all samples, and luminescence signals from drug-treated conditions were normalized to DMSO, which was set to 100%.

### 2.5. Mass Spectrometry

Lipidomic analysis was conducted by electrospray ionization-tandem mass spectrometry (LC/ESI-MS/MS) as previously described [15]. Briefly, total lipids were extracted from cells (400 µg protein), with internal standards spiked in, using ethyl acetate/isopropanol/water (60:30:10, *v*/*v*) without phase partitioning, and solvents were evaporated under a stream of nitrogen. Lipid extract separation was achieved using a Waters I-class Acquity LC and Waters Xevo TQ-S instruments. Results are presented as pmol lipid/mg cell protein, except for dihydro-ceramides (fold-change), due to the low concentration of this ceramide. SACLAC analysis was also performed by LC-MS/MS. C2-dihydroceramide (5 pmol) was used as an internal standard, and an external calibration curve was created for the quantification of SACLAC.

### 2.6. Cellular Respirometry

High-resolution O_2_ consumption measurements were conducted using the Oroboros Oxygraph-2K (Oroboros Instruments, Innsbruck, Austria) in digitonin-permeabilized cells. All experiments were carried out at 37 °C in a 1 mL reaction volume. For intact cell experiments, cells were suspended in bicarbonate-free RPMI-1640, supplemented with 20 mM HEPES, 10% FBS, 100 units/mL penicillin, and 100 µg/mL streptomycin. Respiration was assessed under basal conditions, as well as following the addition of oligomycin (20 nM), FCCP (0.5–3 µM), and rotenone/antimycin (0.5 µM/0.5 µM). For permeabilized cell experiments, cells were centrifuged at 1000 rpm for 7 min at room temperature, washed in respiration buffer, centrifuged once more, and then suspended in respiration buffer at a cell concentration of 2–3 × 10^6^ viable cells/mL. Respiration buffer consisted of potassium-MES (105 mM; pH 7.2), KCl (30 mM), KH_2_PO_4_ (10 mM), MgCl_2_ (5 mM), EGTA (1 mM), and BSA (2.5 g/L). After recording basal respiration, cells were permeabilized with digitonin (20 µg/mL), energized with various carbon substrates (pyruvate, malate, glutamate, succinate, octanoyl-l-carnitine; P, M, G, S, O; 5 mM, 2 mM, 5 mM, 5 mM, 0.2 mM), and flux was stimulated across a physiological ATP free energy demand using the creatine kinase (CK) clamp [16]. The CK clamp system leverages the enzymatic activity of CK to couple the interconversion of ATP and ADP to that of phosphocreatine (PCr) and free creatine (Cr). In this way, this technique allows the extramitochondrial ATP free energy (i.e., ΔG_ATP_) to be clamped at an empirically defined value and then further titrated with sequential PCr additions [17]. Thus, stimulation of mitochondrial respiration can proceed across a physiological ΔG_ATP_ span (i.e., −56 to −64 kJ/mol) [18]. In order to assess the integrity of the outer mitochondrial membrane, respiration was assessed in digitonin-permeabilized cells in the absence and presence of exogenous cytochrome C (10 µM). An increase in respiration in response to cytochrome C is indicative of outer-membrane disruption. Data were normalized to viable cell count and expressed as pmol/s/million cells. All additions were made directly to the O2K chamber during the period of each assay. The typical assay length was 20–40 min.

### 2.7. Immunoblotting

Immunoblotting was conducted as previously described [14,19] with modifications. Employed from Cell Signaling Technologies were phospho-Akt (#4060, phospho-GSK-3β (#9323), Mcl-1 (#5453), caspase-2 (#2224), survivin (#2808), cleaved caspase-9 (#7237), cleaved caspase-3 (#9664) and β-Actin (#8457, 3700). Briefly, cells were lysed in 1× RIPA buffer containing phosphatase inhibitor cocktail 2, protease inhibitor, and PMSF (1.0 mM), according to the manufacturer’s protocol (Cell Signaling Technology, 3 Trask Lane Danvers, MA 01923 USA). The bicinchoninic acid (BCA) method was employed for determining protein concentrations. Lysates were centrifuged to remove debris and then heated in 4× Laemmli sample buffer with 10% b-mercaptoethanol (Bio-Rad, 2000 Market St Suite 1460, Philadelphia, PA 19103 USA) at 95 °C for 5 min. Equal amounts of total protein were separated by SDS–polyacrylamide gel electrophoresis and subsequently transferred onto nitrocellulose membranes (Bio-Rad). The membranes were blocked with 5% nonfat milk in PBS for 1 h at room temperature and incubated with primary antibodies according to the manufacturer’s instructions at 4 °C overnight. The membranes were washed with PBST, followed by incubation with the corresponding HRP-conjugated secondary antibodies according to the manufacturer’s instructions, for 1 h at room temperature. The membranes were washed with PBST before exposure. Bound antibodies were measured using an enhanced chemiluminescence detection kit (Thermo Fisher Scientific) and surveyed using a ChemiDoc imaging system (Bio-Rad). The relative intensity of protein expression was calculated using ImageJ software (RRID:SCR_003070).

### 2.8. Rhodamine Assay

Rhodamine assay, a functional test for P-gp efflux activity, was performed as described previously [20], with slight modification. HL-60/DNR cells (2 × 10^6^ cells/mL complete media) were incubated at 37 °C with the antagonist for 30 min, in a total volume of 1.0 mL, prior to the addition of 0.1 mg rhodamine 123 (dissolved in DMSO) for 1 h. After centrifugation and washing, cell pellets were dissolved in 0.1 mL 0.02% SDS in water, and fluorescence was measured in a microplate reader. 

### 2.9. Flow Cytometry

HL-60/DNR cells (2 × 10^5^/mL RPMI complete medium) were seeded in 12-well plates and treated with indicated agents for 24 h. Cells, plus a 2 mL well wash with ice-cold PBS, were then collected in 5 mL round bottom plastic tubes and centrifuged at 400× *g* for 7 min at 4 °C. The supernatant was discarded, cells were resuspended in 0.1 mL Annexin V binding buffer, and 0.3 µL FITC conjugated Annexin V was added. Cells were incubated for 15 min at room temperature in the dark. After the addition of 0.2 mL of the same binding buffer, DAPI was added to reach a final concentration of 40 ng/mL, and apoptosis was evaluated by Cytek Aurora flow cytometry [21].

### 2.10. Tumor Xenograft Model

HL-60/DNR cells were transduced with pFU-LUC2-eGFP lentivector and sorted for EGFP. pFU-LUC2-eGFP plasmid was gifted by Drs. Sanjiv Sam Gambhir and Tzuhua Dennis Lin (Stanford University School of Medicine). The HL-60/DNR-Luc2-EGFP cells were then cultured in RPMI-1640 media supplemented with 1 µg/mL of DNR (~1.77µM). The exponentially growing cells (~0.6–0.8 × 10^6^ cells/mL) were treated with either 5 µM SACLAC, 10 µM D-*threo*-PDMP, or the mix for 12 h. Cells were then washed and suspended in Hank’s balanced salt solution prior to injection into 7–9 wk old male NOD.Cg-*Rag1^tm1Mom^ Il2rg^tm1Wjl^* Tg (CMV-IL3,CSF2,KITLG) 1Eav/J (NRG-S) mice (Jackson Laboratory, 600 Main Street, Bar Harbor, ME 04609, USA) subcutaneously (2.5 × 10^6^ viable cells/site/animal). Animals were imaged using the IVIS Lumina LT Series III imaging system (Perkin Elmer, Waltham, MA, USA) after 7–10 min of D-luciferin (Gold Biotechnology, 1328 Ashby Rd, St. Louis, MO 63132, USA) injection (150 mg/kg; IP) on indicated days. The photons emitted from cells expressed as total flux (photons/s/cm^2^/steradian) were quantified and analyzed using Living Image software 4.8 (Perkin Elmer). Tumor volume = [π × length (mm) × width (mm)^2^]/6, where length represents the largest tumor diameter, and width represents the perpendicular tumor diameter. GraphPad Prism version 6.0 (RRID:SCR_002798) was used to compute the statistical analysis and plot the data.

### 2.11. Statistical Analysis

Mitochondrial assay results are expressed as the mean ± SEM (error bars). Throughout the paper, differences between groups were assessed by *t*-test, or one-way ANOVA, followed by Tukey’s test where appropriate using GraphPad Prism 8 software (Version 8.4.2) (RRID:SCR_002798). Other statistical tests used are described in the figure legends. Statistical significance in the figures is indicated as follows: * *p* < 0.05; ** *p* < 0.01; *** *p* < 0.001; **** *p* < 0.0001. Unless otherwise stated, figures were generated using GraphPad Prism 8 software (Version 8.4.2) (RRID:SCR_002798).

## 3. Results

### 3.1. Response to Chemotherapy Challenge in HL-60 Wild-Type and in Drug-Resistant VCR- and DNR-Selected Counterparts

As a means to verify drug resistance in our human AML cell models, studies were conducted in HL-60 wt and in the VCR- and DNR-resistant counterparts. Further, to determine if drug resistance would diminish after culture in a drug-free medium, VCR- and DNR-resistant cells were grown without drugs for 6-wk (designated 6-wk) and then evaluated. HL-60/VCR and HL-60/VCR6-wk cells were resistant to VCR at concentrations up to 10 µg/mL (Figure 1A, left), whereas viability in wt cells fell to 35% at 0.05 µg/mL (Figure 1A, right, note the different *x*-axis). Drug resistance was also a strong characteristic feature in HL-60/DNR cells (Figure 1B). For example, at 1.0 ug/mL DNR, HL-60/DNR, and HL-60/DNR6-wk cells maintained 100% viability, whereas in wt cells, viability fell to 5%. Drug resistance was also maintained at higher DNR concentrations in the HL-60/DNR model. We have previously shown that AC and GCS upregulation accompanied drug resistance elicited by chemotherapy selection pressure [5,15], and thus simultaneously blocking these ceramide-regulating routes might be of interest from a therapeutic standpoint in the drug resistance setting.

### 3.2. Impact of SACLAC, D-threo-PDMP, and Mix on Viability in Drug-Resistant and Wild-Type Human AML Cell Lines

In order to determine whether blocking AC or GCS would be cytotoxic, drug-resistant HL-60 cells and other representative wt human AML cell lines were exposed to either SACLAC or D-*threo*-PDMP. Qualitatively, all cell lines exhibited similar insensitivity to D-*threo*-PDMP (Figure 2A); consequently, all displayed comparative, qualitative sensitivity to SACLAC. These responses suggest that AC activity may be a predominant ceramide-eliminating avenue in the AML. We next determined the effect of co-administering SL enzyme inhibitors to simultaneously block ceramide hydrolysis and glycosylation. Coadministration of D-*threo*-PDMP and SACLAC was synergistic in drug-resistant AML cells (VCR and DNR models), compared to HL-60 wt and other wt human AML cell lines not of HL-60 origin (MV4-11, MOLM-13, and MOLM-14) where combination benefit was absent (Figure 2B). For example, in HL-60/VCR cells, SACLAC and D-*threo*-PDMP imparted respective 10 and 0% reductions in viability at 24 h, whereas the combination reduced cell viability to 50%. To verify the efficacy of D-*threo*-PDMP and SACLAC in drug-resistant cells, we used trypan blue exclusion and cell counting. As shown in Figure 2C, the inhibitor combination was clearly more effective than single agents. For example, in HL-60/DNR cells, whereas SACLAC and D-*threo*-PDMP were alone minimally effective, the combination reduced viability to 48 and 2% at 24 and 48 h, respectively. Of note, the unusual MOLM 14 response, stimulatory with the combination, is perhaps related to the generation of mitogenic lipids such as sphingosine 1-phosphate. In addition, we tested the inhibitor mix in KG-1a (Figure 2C), another P-gp-expressing human AML cell line [22]. Whereas D-*threo*-PDMP was not cytotoxic and SACLAC-treated cells were 76% viable, the mix produced 47% viability. It is likely that complete cell kill was not achieved in some instances because inhibitor concentrations were too low and/or the result of other survival mechanisms such as DNA damage repair, epigenetic changes, and inhibitor inactivation. These data underscore the cytotoxic impact of this SL enzyme inhibitor combination in the drug-resistant setting. In PBMCs, treatment was only moderately cytotoxic at 24 h (88% cell viability); viability was maintained at 77% at 48 h (Figure 2D). Lastly, the data in Figure 2E verify the presence of P-gp, a hallmark of multidrug resistance, in DNR- and VCR-resistant cells, as well as KG-1a cells, albeit low, compared to wt AML cell lines.

### 3.3. Lipidomics—Impact of SACLAC and D-threo-PDMP on Sphingolipid Profiles, a Recipe for Cytotoxicity

Initially, we sought to determine the effects of DNR resistance on cellular ceramide levels, and to this end, Figure 3A shows that HL-60/DNR cells displayed a preponderance of ceramides, specifically the C16:0, C24:1, and C26:1 molecular species, compared to HL-60 wt cells. Of note, however, the proliferative, anti-apoptotic very-long-chain ceramide species [23,24]: C24:1 and C26:1, were greatly elevated in HL-60/DNR cells compared to wild-type, whereas pro-apoptotic long-chain C16:0 ceramide increased to a lesser extent in DNR cells. This shift resulted in a reduction in the C16:0/C24:1 ratio from 0.627 ± 0.03 (*n* = 5) to 0.30 ± 0.007 (*n* = 3), a pro-survival bent in HL-60/DNR cells. With reference to the blockade of ceramide metabolism, we anticipated that lipidomic analysis in cells exposed to SACLAC and D-*threo*-PDMP would allow for the understanding of how SL changes support the observed cytotoxic responses. AC inhibition by SACLAC produced striking, multi-fold increases in numerous ceramide molecular species in HL-60/DNR cells (Figure 3B). For example, by impeding intracellular ceramide hydrolysis, SACLAC increased the levels of C16, 18, 20, 22, 22:1, 24, 24:1, and 26:1 ceramide by 4-, 21-, 18-, 8-, 5-, 2-, 3-, and 1.5-fold, respectively. D-*threo*-PDMP exposure also increased ceramide levels over control (C16, C22, C24, C24:1), suggesting that glycosylation, to a lesser extent, is also a mode of ceramide elimination in drug-resistant HL-60/DNR cells. D-*threo*-PDMP exposure plummeted levels of all major glucosylceramide (GC) molecular species (Figure 3C), indicative of very effective GCS inhibition. The near-total block in ceramide glycosylation caused by D-*threo*-PDMP significantly augmented ceramide levels in cells when SACLAC was added (Figure 3B), such that fold-increases in C16, C18, C20, C22, C22:1, C24, C24:1, and C26:1 jumped from 4 to 7-fold, 21 to 28-fold, 18 to 26-fold, 8 to 11-fold, 5 to 10-fold, 2 to 6-fold, 3 to 6-fold, and 1.5 to 3-fold, in SACLAC versus combination treatment, respectively. This augmentation is perhaps driving the heightened reduction in cell viability demonstrated with combination treatment. Exposure to SACLAC increased the levels of several molecular species of GC (C16, C18, C20, C22, C22:1, C24:1, C26:1) (Figure 3C), presumably a “cell-saving” response to the ceramide surplus caused by SACLAC blocking ceramide hydrolysis. The surge in ceramide levels was also accompanied by increases in dihydro (dh)-ceramides with combination exposure (Figure 3D). Analysis of long-chain sphingolipid bases revealed marked changes in levels of these bioactive SLs. Levels of sphingosine, the S1P precursor, and S1P, exhibited precipitous decline with SACLAC exposure (Figure 3E), which is to be expected when ceramide hydrolysis is prevented. The enzyme inhibitor combination, which synergistically reduced cell viability, did effectively reduce S1P levels by >50% of the control value. D-*threo*-PDMP increased S1P levels, perhaps by making more ceramide available (Figure 3B). The increase in dh-Sphingosine levels with combination enzyme inhibitor treatment is likely due to increased dh-ceramides, resulting from the flood of ceramides generated. D-*threo*-PDMP exposure produced a nearly 30% drop in C16 SM content (Figure 3F). However, increases in several SM molecular species occurred, notably C16, C18, C22, C24, and C24:1, in cells exposed to the combination. This maneuver perhaps serves as a sink to shield cells from the buildup of free ceramides, although too much SM could do damage. Interestingly, exposure to the enzyme inhibitors also increased ceramide levels in HL-60 wt cells (Figure 3G). Although of lesser measure in wt cells, the patterns and fold-increases of ceramide species generated were similar with HL-60/DNR cells.

### 3.4. Dissecting the Role of P-gp in Manipulating Ceramide Metabolism in the SACLAC D-threo-PDMP Regimen 

It is well established that inhibition of P-gp function can limit ceramide glycosylation, and because the inhibitor regimen demonstrated efficacy in drug-resistant, P-gp-expressing AML models, we sought to assess the role of P-gp in this process. To investigate this possibility, we utilized D,L-*erythro*-PDMP, an inactive isomer of D-*threo*-PDMP [25], that does not inhibit GCS but can limit ceramide glycosylation via modulation of P-gp function [7,8,26]. Results show that the inactive form of PDMP, the D,L-*threo*, while not cytotoxic alone, heightened the SACLAC effect in HL-60/DNR cells (Figure 4A, far right). For example, viability was nearly 80% with 5 µM SACLAC, 100% with 10 µM D,L-*erythro*-PDMP, and 32% with the combination, a clear enhancement. Similarly, in HL-60/VCR cells, viability was 80% with SACLAC exposure, 102% with D,L-*erythro*-PDMP exposure, and 38% with the combination (investigators laboratory). To further position the participation of P-gp as a player, we tested cyclosporin A, a commonly employed ABC transporter antagonist [27,28], verapamil, another classic P-gp antagonist, and zosuquidar, a high-affinity P-gp antagonist. The data in Figure 4A show that although all were nontoxic alone, all were beneficially coactive when partnered with SACLAC. Thus, combinatorial SACLAC plus P-gp antagonist phenocopied the synergistic cytotoxicity imparted by SACLAC and D-*threo*-PDMP. Importantly, in P-gp naïve HL-60 cells, neither D,L-*erythro*-PDMP nor any of the P-gp antagonists showed enhanced cytotoxicity when partnered with SACLAC (Figure 4B). As an example, SACLAC plus verapamil yielded a cytotoxic response identical to SACLAC alone. Of note, some of the agents yielded significant differences when partnered with SACLAC compared to single agents; however, the trend was in the anti-apoptotic direction. These data suggest that P-gp participates in driving ceramide-orchestrated cytotoxicity imparted by the combination of SACLAC and D-*threo*-PDMP. Interestingly, we assessed the clinically approved, highly specific inhibitor of GCS, eliglustat [29], effective at nM concentrations [30], with SACLAC in HL-60/DNR cells and found poor combination benefit at both 1.0 and 10 µM (Figure 4C). In order to determine if this lack of activity was due to poor affinity of eliglustat for P-gp, eliglustat was evaluated in rhodamine trapping assays. Rhodamine, a P-gp substrate, is actively pumped by P-gp; this action is blocked in the presence of P-gp antagonists. As shown in Figure 4D, intracellular rhodamine fluorescence was greatly enhanced by verapamil, serving as a positive control, and by D-*threo*-PDMP; however, the effect of eliglustat was without consequence, suggesting that, unlike D-*threo*-PDMP, eliglustat does not block P-gp function in HL-60/DNR cells. In summary, these studies demonstrate that P-gp is a prime target for countering ceramide glycosylation to enhance the “ceramide response”.

In view of the synergy between SACLAC and D-threo-PDMP in drug-resistant cells, we sought to determine whether SACLAC was pumped and whether PDMP blocked this action, a scenario that would maintain high intracellular concentrations of this AC inhibitor. To test this, we exposed HL-60/VCR cells to 10 µM SACLAC for 3 h, in the absence and presence of either cyclosporin A, D-*threo*-PDMP, or D,L-*erythro*-PDMP (all at 10 µM) and determined intracellular levels of SACLAC (by mass spectroscopy). None of these treatments had a statistically significant impact on the levels of intracellular SACLAC (investigators laboratory), showing that SACLAC is not a P-gp substrate and that SACLAC trapping is not part of the cytotoxic response.

### 3.5. Mechanisms Underlying Cytotoxicity—Signaling Events and Mitochondrial Perturbation

Signaling events underlying cytotoxic responses to the SL enzyme inhibitor combination were evaluated first by employing assays to mark possible caspase involvement. As shown in Figure 5A, treating either HL-60/DNR or HL-60/VCR cells in combination with SACLAC and D-*threo*-PDMP (1:2 molar ratio) resulted in dose-dependent increases in caspase-3/7 activity. Of note, background constitutive caspase activity was three-fold higher in HL-60/DNR cells compared to the VCR-resistant model. Confirmation of caspase activation was made by immunoblot assays; time-dependent increases (4–24 h) in cleaved initiator caspase-9 and cleaved executioner caspase-3 occurred with exposure to the SACLAC D-*threo*-PDMP regimen (Figure 5B, bottom). Cleavage of caspase-2, an initiator caspase upstream of mitochondria, was also demonstrated (Figure 5B, top left). In addition, exposure to the inhibitor mix promoted inhibition of Akt activation, shown by decreased pAkt levels, and activation of GSK-3β, shown by decreased levels of pGSK-3β, connoting GSK-3β involvement in ceramide-induced mitochondrial apoptosis. Increased expression of pro-apoptotic Mcl-1S, the splice variant of anti-apoptotic Mcl-1L, the predominant form in many cancers, was also shown (Figure 5B, left). Lastly, we demonstrate a small but significant decrease in survivin expression upon exposure to the mix. Protein quantitation is shown in Figure 5B, right. Flow cytometry using Annexin V binding was used to affirm cellular apoptosis. As in the caspase assay (Figure 5A), control HL-60/DNR cells were stressed, as evidenced by the high background in early and late apoptosis (Figure 5C, Control). When the background is taken into account, the results demonstrate that whereas single agents imparted slight, if any, increases in Annexin V binding (early and late apoptosis), the combination (Mix) synergistically increased apoptosis after only a 24 h exposure. The intrinsic pathway of apoptosis intervenes at mitochondria and impacts mitochondrial function. An additional mechanism of action led us to interrogate mitochondrial bioenergetics, especially considering the titanic increases in ceramides resulting from SACLAC D-*threo*-PDMP exposure. To evaluate the impact of the drug combination on mitochondrial respiration, both HL60 wt and HL60/DNR were exposed to the drug combination. Following a 24 h exposure, mitochondrial respiratory kinetics were assessed in both intact and digitonin-permeabilized cells (Figure 5D,E). Relative to DMSO control, both basal and maximal respiration in intact cells was decreased by the inhibitor mix in HL60/DNR cells (Figure 5D). Interestingly, upon digitonin permeabilization, HL60/DNR cells treated with the mix were completely refractory to respiratory stimulation (Figure 5E). Specifically, despite no change in respiration in the presence of either digitonin alone or saturating pyruvate/malate, direct stimulation of oxidative phosphorylation (OXPHOS) revealed a complete inability of HL60/DNR cells to respond to the OXPHOS stimuli (Figure 5E). Lowered permeabilized cell respiration in response to SACLAC D-*threo*-PDMP was independent of both the carbon substrate supplied (i.e., similar OXPHOS deficiency in the presence of pyruvate/malate, glutamate/octanoyl-carnitine, and succinate), as well as the ATP resynthesis demand state (Figure 5E). In permeabilized cells assays, exogenous cytochrome C increased respiration only in HL60/DNR exposed to the combination of SACLAC D-*threo*-PDMP, consistent with outer-membrane loss of integrity (Figure 5F). Interestingly, the SACLAC D-*threo*-PDMP drug combination had minimal effects in HL60 wt cells, consistent with our observations of maintained viability in HL60 wt cells upon administration of the SACLAC D-*threo*-PDMP combination. 

### 3.6. In Vivo Efficacy in an Animal Model and Activity in Patient-Derived AML Cells

To test the effect of SL enzyme inhibitors in vivo, we chose a xenograft model using HL-60/DNR cells (HL-60/DNR-Luc2-EGFP) that were pretreated before implantation (Figure 6A). This model was preferred as SACLAC is, in our experience, poorly bioavailable, and until delivery issues are resolved, perhaps via nano-formulation, this model provided initial insight. As summarized in Figure 6B–F, cells receiving combination treatment, assessed 17 days post-implantation, produced less radiance (Figure 6B,C), diminished tumor size, weight, and volume, compared to DMSO controls (Figure 6D–F) and compared with exposure to single agents. These results show that the preexposure of HL-60/DNR cells to the inhibitors sets in motion a biochemical cascade that limits tumor growth upon in vivo presentation.

To test efficacy in patient-derived AML cells, a subset of patients was evaluated with SACLAC, D-*threo*-PDMP, or the combination. Approximately 70% demonstrated increased efficacy with the combination over single agents. The patient cell viability data presented represent the top 9 samples showing the most potent combined drug effects over single agents (Figure 7). As shown, the combination benefit from the enzyme inhibitor mix is readily evident in AML patient samples.

## 4. Discussion

This work highlights the importance of targeting SL metabolism for therapeutic application in AML, most notably in chemotherapy-resistant AML, as herein we demonstrate the effectiveness of blocking AC and ceramide glycosylation, pivotal control points in the metabolism of ceramide. The efficacy of SACLAC as a potent AC inhibitor has been presented [5,31,32], and a large literature exists on the GCS inhibitors such as D-*threo*-PDMP, D-*threo*-PPMP, eliglustat, and the like, for treatment of lipid storage diseases as well as cancer [33].

Our lipidomic data allow for an understanding of how SL changes support the observed cytotoxic responses. For example, simultaneous inhibition of AC and ceramide glycosylation yielded higher ceramide levels than either blockade alone. In cancer cells, it is a well-known ploy to shuttle ceramide into GC; this lessens the ceramide burden dulling subsequent ceramide-orchestrated cell death [6,34]. Therefore, in this instance, blocking ceramide glycosylation [35] makes sense. It is noteworthy that levels of all major ceramide molecular species were increased by this regimen. Capital among these were apoptosis-inducing, antiproliferative C16, C18, and C22 species. In a model of head and neck squamous cell carcinoma, large C18 ceramide increases were attributed to cell death in response to gemcitabine/doxorubicin treatment [36]. In leukemia-related studies, C18 ceramide generation has been linked to AML cell death in response to FLT-ITD inhibition [37].

AC is upregulated in AML [14,38], and as such, this enzyme presents an attractive therapeutic target [39,40]. On point, AC regulates the balance between a ceramide’s apoptosis-inducing impact and S1P and its drive to mitogenicity. Thus, AC inhibitors take on a paramount role in controlling the SL rheostat: the balancing act between cell life and death. Here it is important to note that our enzyme inhibitor combination reduced S1P levels in HL-60/DNR cells by 50% compared to controls, another antiproliferative benefit complementing the large ceramide upswings observed with the D-*threo*-PDMP SACLAC regimen.

The Inhibitor regimen was more effective in drug-resistant, P-gp-expressing AML cells as opposed to wt. This calls into question the role of P-gp in enhancing the cytotoxic response to ceramide as some GCS inhibitors, such as D-*threo*-PDMP, interact with P-gp. For example, P-gp has been shown to facilitate ceramide glycosylation, a step that, if antagonized, can enhance ceramide-driven cytotoxicity [8,41]. In accord, Golgi-resident P-gp can function as a GC transmembrane flippase [42], pumping GC into the Golgi and removing product (GC) from the active site of GCS, a step that aids in perpetuating ceramide glycosylation. A work by Shabbits and Mayer [43] showed that P-gp can regulate ceramide-mediated sensitivity to paclitaxel, a ceramide-generating taxane [44], implying that ceramide metabolism and apoptosis are regulated by both GCS and P-gp. Moreover, the work of Morad et al. [8] strongly substantiated the participation of P-gp in ceramide metabolism by showing that P-gp antagonists enhanced ceramide-based therapeutics via the blockade of ceramide glycosylation. Further, the work of Sietsma et al. [45] demonstrated that PDMP decreased paclitaxel and vincristine efflux in neuroblastoma cells, thus acting as a P-gp antagonist, and Plo et al. [22] showed that D-*threo*-PDMP increased rhodamine retention and acted as a chemosensitizer in P-gp expressing AML cells, including KG-1a. These results support our findings and support targeting P-gp to enhance the “ceramide effect”. An excellent review summarizes how the ATP-binding cassette transporters, such as P-gp mediate differential biosynthesis of glycosphingolipid species [46]. We posit that the cytotoxic benefits of the SL enzyme inhibitor regimen in drug-resistant AML are due in part to the interaction of D-*threo*-PDMP with P-gp, acting to block ceramide glycosylation and advancing ceramide-driven apoptosis that is co-assisted via AC inhibition.

Large increases in ceramide, caspase activation, and altered Akt, GSK-3β, and Mcl-1 expression accompanied D-*threo*-PDMP SACLAC-induced cell death and are known factors in ceramide-driven, caspase-regulated apoptosis [12]. Considering the severe hit to respiration (see Figure 5E), the SL enzyme inhibitor regimen appears keenly mitochondria-centric. The damage is likely due to the properties of ceramide to form channels in the mitochondrial outer membrane, eliciting outer membrane permeabilization, a step that corrupts bioenergetic service [47,48]. Recent work demonstrates that drug-resistant AML is hallmarked by intrinsic deficiencies in OXPHOS that limit oxidative metabolism in the presence of ATP-free energy [15,49]. Although speculative, it is possible that the mitochondrial alterations well known to accompany AML drug resistance do not serve to enhance cellular oxidative energy production but rather to endow the cells with an increased ability to thwart apoptosis. In support of this, the inhibitor regimen induced a slight increase in respiration in intact cells; however, clear disruptions in mitochondrial respiratory membrane integrity were apparent upon digitonin permeabilization. Therefore, ceramide-generating therapies, such as our SACLAC D-*threo*-PDMP regimen, presumably induce cell death via circumventing anti-apoptotic defenses afforded by the drug-resistant mitochondrial network. Importantly, pro-apoptotic changes across the mitochondrial network clearly occur prior to any overt changes in cellular oxidative capacity, thus highlighting the diagnostic limitations of intact cell respirometry alone. Based on these data, rather than targeting mitochondrial oxidative metabolism directly, induction of cell death in drug-resistant AML is perhaps more efficacious in the setting of drug combinations that specifically disrupt mitochondria anti-apoptotic defenses.

Akt, a survival signaling factor playing a vital role in cancer development and progression, is often activated in AML [50,51]. Akt represents an important therapeutic locus targeted by our inhibitor regimen. Treatment of HL-60/DNR cells resulted in inhibition of Akt activation (reduction in pAkt). Apropos in the current context, exposure to an Akt kinase inhibitor was shown to deplete pro-survival S1P levels and increase pro-apoptotic ceramide levels in an AML model [50], similar to our findings with the “ceramide-generating” SL enzyme inhibitors. Also in our system, GSK-3β, which is upstream of mitochondrial, stress-induced apoptosis, was activated (via protein dephosphorylated) upon exposure to the SACLAC D-*threo*-PDMP combination. In further support of ceramide-driven signal transduction, Rahman et al. [52] have shown that the antileukemic, apoptotic potential of sanguinarine, an alkaloid isolated from bloodroot, involves ceramide generation, Akt inhibition, and dephosphorylation of downstream GSK-3β, similar with our findings. Lastly, the expression of Mcl-1, a prominent target in cancer therapy [1,53,54,55], a member of the anti-apoptotic BCL-2 family, and a predictor of poor prognosis and disease recurrence in AML [56], was impacted by the inhibitor combination. This was evidenced by upregulated expression of Mcl-1S, the short isoform of Mcl-1, and the driver of pro-apoptotic responses [57,58]. These data support work of Pearson et al. [31], who demonstrated that SACLAC treatment led to reduced levels of splicing factor SF3B1 and alternative Mcl-1 mRNA splicing that increased Mcl-1S levels and contributed to SACLAC-induced apoptosis in AML cells.

From a therapeutic standpoint, although the in vivo model chosen was not ideal, we await new, novel formulations of SACLAC that improve stability and bioavailability to be used in concert with D-*threo*-PDMP or with P-gp antagonists, such as verapamil, cyclosporin A, tamoxifen, or the highly specific antagonist, zosuquidar. These data underscore the therapeutic benefit of targeting ceramide metabolism at P-gp as opposed to GCS [34,59]. Dual targeting of metabolic control points that regulate intracellular ceramide levels represents a promising direction in the treatment of drug-resistant AML and perhaps in the treatment of drug-resistant solid tumors.

## 5. Conclusions

Upregulated ceramide metabolism appears to be a hallmark of drug-resistant cancers. Such upregulation, manifested by increases in GCS, AC, and SPHK1 expression, join to benefit cancer cell survival and GCS/P-gp and AC activities contributing to ceramide clearance, while SPHK1, famous for producing sphingosine 1-phosphate, is a mitogenic, anti-apoptotic player. To capture the anticancer, pro-apoptotic elements of ceramide, we employed dual blockade of ceramide hydrolysis and glycosylation, a track that resulted in titanic increases in intracellular ceramide levels, a shift in the ratio of long-chain ceramides to very long-chain ceramides, mitochondrial damage, and cell death. The blockade was synergistic in P-gp-rich, drug-resistant models but not in P-gp-poor wild-type cells. Because P-gp antagonists were shown to replace D-*threo*-PDMP in the SACLAC regimen, we conclude that targeting P-gp, as opposed to GCS specifically, may be a more favorable route for limiting ceramide glycosylation to enhance “the ceramide effect” in treatment of drug-resistant cancer.

## Figures and Tables

**Figure 1 cancers-15-01883-f001:**
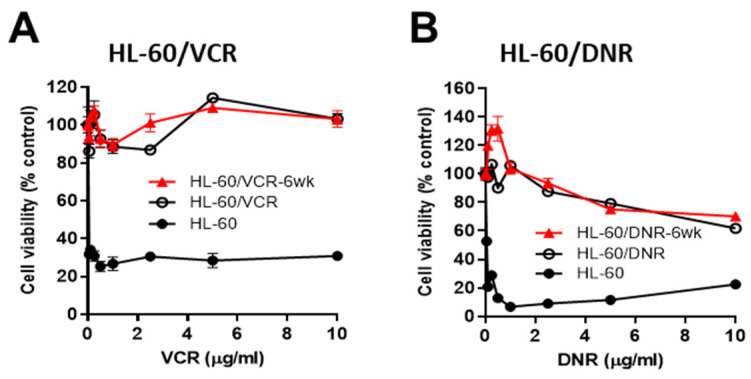
Chemotherapy sensitivity in HL-60 wild-type and in VCR- and DNR-selected counterparts. (**A**,**B**), Cells (50,000/well) were seeded into 96-well plates in complete media and exposed to VCR or DNR at the concentrations indicated for 72 h (drug-resistant cells) and 48 h (HL-60 wt cells), after which viability was determined using the MTS assay. The red line indicates chemotherapy sensitivity in HL-60/VCR and HL-60/DNR cells that were cultured in VCR- and DNR-free media for 6-wk prior to assay. Data represent Mean ± SEM, (**A**,**B**) N = 6/group.

**Figure 2 cancers-15-01883-f002:**
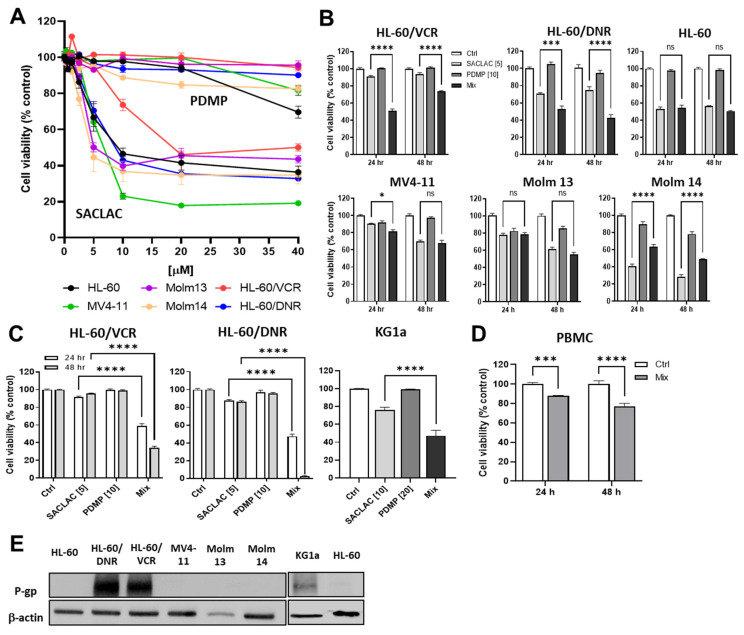
Impact of SACLAC and D-*threo*-PDMP exposure on viability in drug-resistant and wild-type human AML cell lines. (**A**) Impact of single agent exposure. Cells (50,000/well) were seeded into 96-well plates and exposed to either SACLAC or D-*threo*-PDMP for 24 h at the concentrations indicated. Cell viability was determined by PI assay. (**B**) Effect of single agent and mix regimens on AML cell viability. Exposure times and concentrations as indicated. Viability by PI assay. (**C**) Confirmation of combinatorial efficacy in HL-60/VCR, HL-60/DNR, and KG-1a cells by cell counting method. Cells (200,000/well) were seeded into 12-well plates in complete media and treated as indicated for the times shown (KG-1a, 24 hr treatment), after which viability was determined by trypan blue exclusion and cell counting. (**D**) Enzyme inhibitor sensitivity in PBMCs. PBMCs (200,000 cells/well) were seeded into 12-well plates in complete media and exposed to the agents indicated (5 µM SACLAC and 10 µM D-*threo*-PDMP) for the times shown, after which cell viability was determined by trypan blue exclusion and cell counting. (**E**) Western blot analysis of P-gp expression in AML cell lines. Numbers in brackets indicate µM concentrations. Mix refers to SACLAC + PDMP at a 1:2 ratio. PDMP refers to D-*threo*-PDMP. PBMC, peripheral blood mononuclear cells. The uncropped bolts are shown in Appendix A. Data represent Mean ± SEM, (**A**) N = 6/group, (**B**) N = 6/group, (**C**) N = 3–6/group, (**D**) N = 3–6/group. Two-way ANOVA (Tukey’s multiple comparisons test) was used for statistical analysis (* *p* < 0.05, *** *p* < 0.001, **** *p* < 0.0001). ns, not significant.

**Figure 3 cancers-15-01883-f003:**
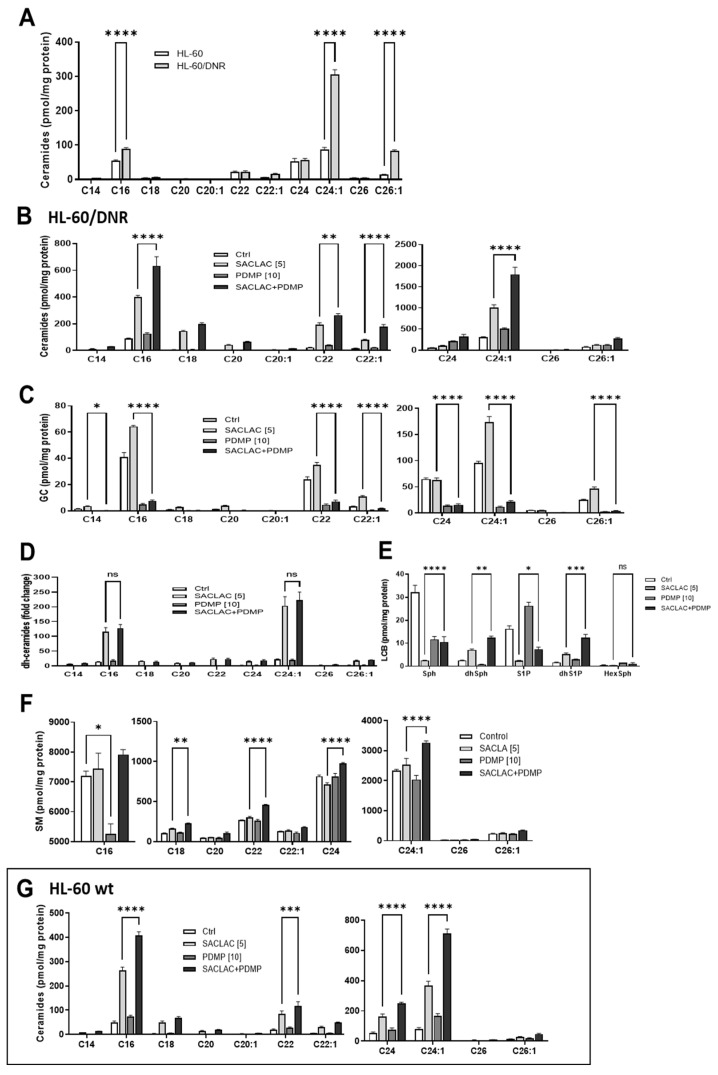
Ceramide levels in HL-60 wt and HL-60/DNR cells and the impact of SACLAC, D-*threo*-PDMP, and the combination regimen on sphingolipid composition. (**A**) Ceramide levels in HL-60 wt and drug-resistant HL-60/DNR cells. (**B**–**F**) Impact of SACLAC, D-*threo*-PDMP, and combination in HL-60/DNR cells on levels of (**B**) Ceramide; (**C**) Glucosylceramide; (**D**) dihydro-Ceramide; (**E**) Long-chain bases; (**F**) Sphingomyelin; and (**G**) Ceramide molecular species in HL-60 wt cells treated with SACLAC, D-*threo*-PDMP, and combination. Cells (6 × 10^6^/10 cm dish in complete media) were exposed to the control (vehicle, DMSO), SACLAC, D-threo-PDMP, or the combination regimen for 24 h, after which cells were harvested, pelleted by centrifugation, washed 3× in cold phosphate-buffered saline, and stored at −80 °C. Lipidomic analysis was conducted by mass spectroscopy as indicated in Methods. Results shown are the average of *n* = 4 experiments. The dh-ceramides levels are given as fold-change due to their low concentrations. GC, glucosylceramide; dh, dihydro; LCB, long-chain base; SM, sphingomyelin; and PDMP, D-*threo*-PDMP. Data represent Mean ± SEM, (**A**) N = 3–5/group, (**B**–**F**) N = 3/group, (**G**) N = 6/group. Two-way ANOVA (Tukey’s multiple comparisons test) was used for statistical analysis (* *p* < 0.05, ** *p* < 0.01, *** *p* < 0.001, **** *p* < 0.0001). ns, not significant.

**Figure 4 cancers-15-01883-f004:**
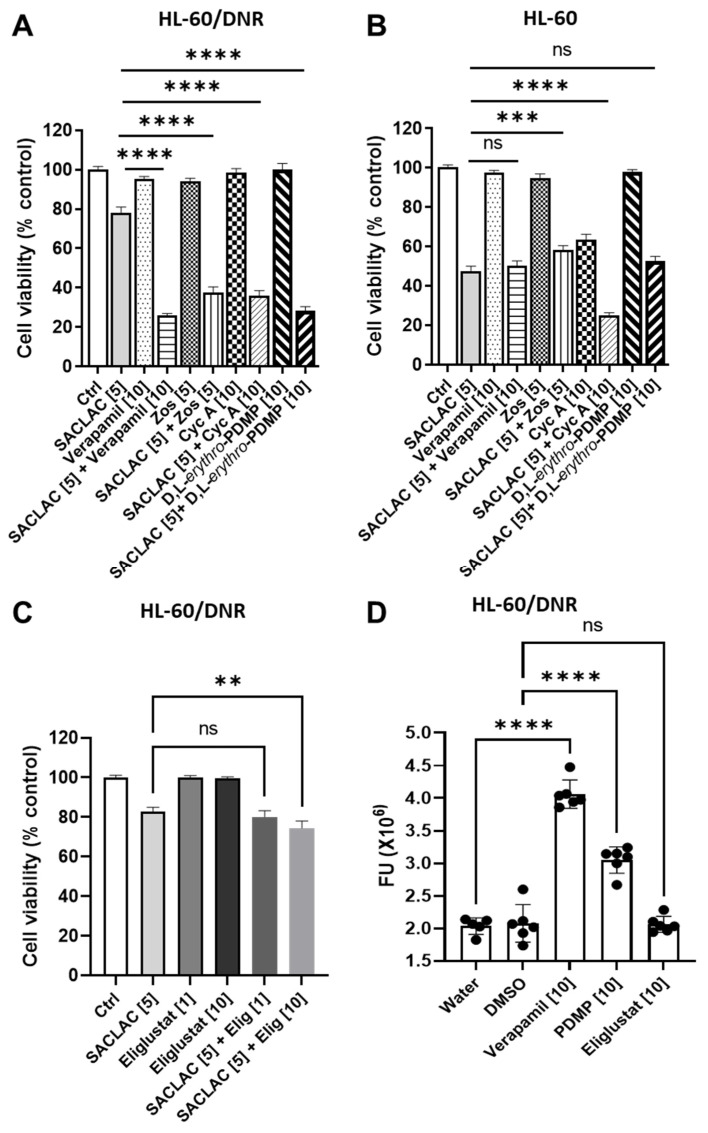
P-glycoprotein antagonists and inactive PDMP isomers enhance SACLAC response in drug-resistant AML cells. (**A**,**B**) Effect of P-gp antagonists and inactive PDMP isomers, D,L-*erythro*-PDMP, on SACLAC cytotoxicity HL-60/DNR (**A**) and HL-60 wt cells (**B**) Cells (50,000/well) seeded in 96-well plates in complete media were treated with SACLAC minus and plus designated agents. After 24 h, viability was determined by the PI assay. (**C**) Effect of SACLAC, the specific GCS inhibitor, eliglustat, and combination on HL-60/DNR cell viability. Cells were exposed to agents for 24 h, and viability was by PI assay. (**D**) Impact of P-gp antagonist, Verapamil, and GCS inhibitors on rhodamine efflux in HL-60/DNR cells. Cells were exposed to agents for 30 min before the addition of rhodamine for 1 h. Zos, zosuquidar; Cyc A, cyclosporin A; Elig, eliglustat; FU, fluorescence units. PDMP in panel D refers to D-*threo*-PDMP. Concentrations in brackets [µM]. Data represent Mean ± SEM, (**A**) N = 6–18/group, (**B**) N = 12/group, (**C**) N =16–18/group, (**D**) N = 5–6/group. Two-way ANOVA (Tukey’s multiple comparisons test) was used for statistical analysis (** *p* < 0.01, *** *p* < 0.001, **** *p* < 0.0001, ns means non-significant).

**Figure 5 cancers-15-01883-f005:**
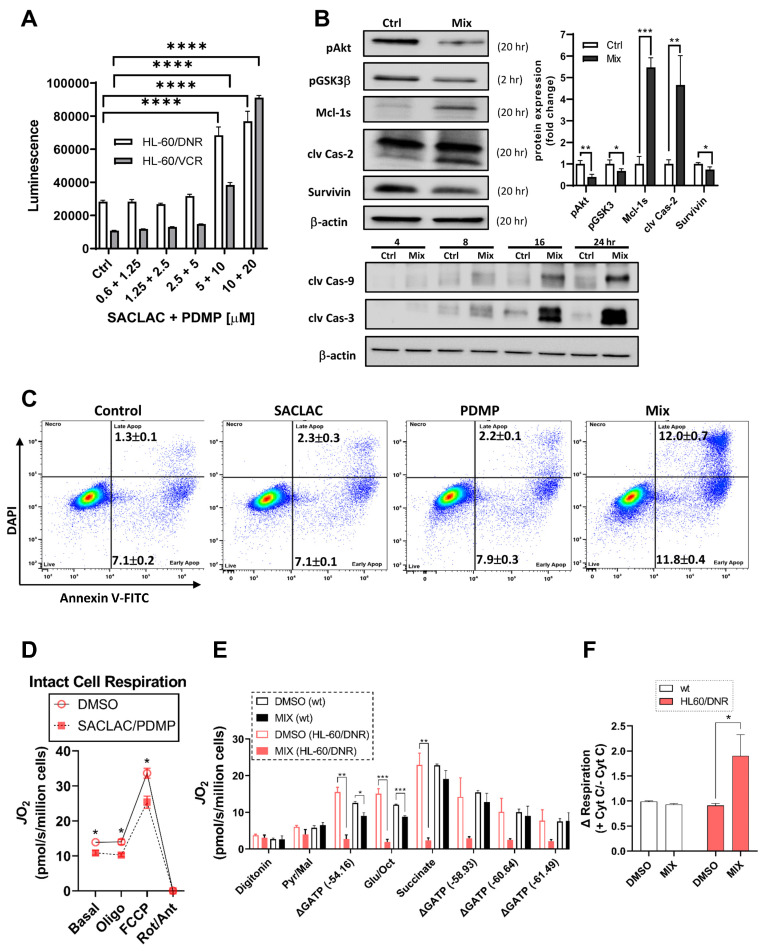
Molecular signals and mitochondria effects supporting cellular responses to SACLAC D-*threo*-PDMP coadministration. (**A**) Caspase 3/7 activation in drug-resistant cells in response to combination SACLAC D-*threo*-PDMP dose escalation (numbers on the *x*-axis refer to SACLAC plus D-threo-PDMP, µM concentrations. (**B**) Top left and bottom, Western blot determination of cleaved caspases 2, 9, and 3, and levels of pAkt, pGSK-3β, Mcl-1S, and survivin, in control and mix (SACLAC + D-*threo*-PDMP, 5 + 10 µM, except pGSK-3β was 10 + 20) treated HL-60/DNR cells. Numbers in parenthesis refer to treatment times, h. (**B**) Top right, Western blot quantitation of designated signaling elements in control and mix-treated HL-60/DNR cells. The uncropped bolts are shown in Appendix A. (**C**) Flow cytometric analysis of HL-60/DNR cells after 24 h exposure to SACLAC, D-*threo*-PDMP, and mix (SACLAC, 5 µM; D-*threo*-PDMP, 10 µM). Numbers refer to the percentages of early and late apoptosis; average ± SEM from three separate experiments. (**D**) Intact cellular respiration in HL-60/DNR cells after 24 h exposure to SL enzyme inhibitor mix (5 + 10 µM). (**E**) Mitochondrial respiratory kinetics in HL-60 wt and HL-60/DNR cells after 24 h exposure to SL enzyme inhibitor mix (5 + 10 µM). (**F**) Mitochondrial respiration in digitonin-permeabilized cells in response to exogenous cytochrome C (10 µM). Cells were energized with saturating carbon (pyruvate, malate, succinate, glutamate, and octanoyl-L-carnitine). Data expressed as a fold change in respiration induced by cytochrome C. Mitochondrial respiratory kinetics were assessed using high-resolution O_2_ consumption, as detailed in Methods. Caspase 3/7 activation was determined using Caspase Glo, from Promega, according to the manufacturer’s instructions. Immunoblotting was conducted as detailed in Methods. Data represent Mean ± SEM, (**A**) N = 6/group, (**B**) N = 3–4/group, (**C**) N = 3/group, and (**D**–**F**) N = 3–4/group. (**A**–**C**) Two-way ANOVA (Tukey’s multiple comparisons test) was used for statistical analysis (* *p* < 0.05, ** *p* < 0.01, *** *p* < 0.001, **** *p* < 0.0001). (**D**–**F**) *t*-test was used for statistical analysis (* *p* <0.05, ** *p* < 0.01, *** *p* < 0.001).

**Figure 6 cancers-15-01883-f006:**
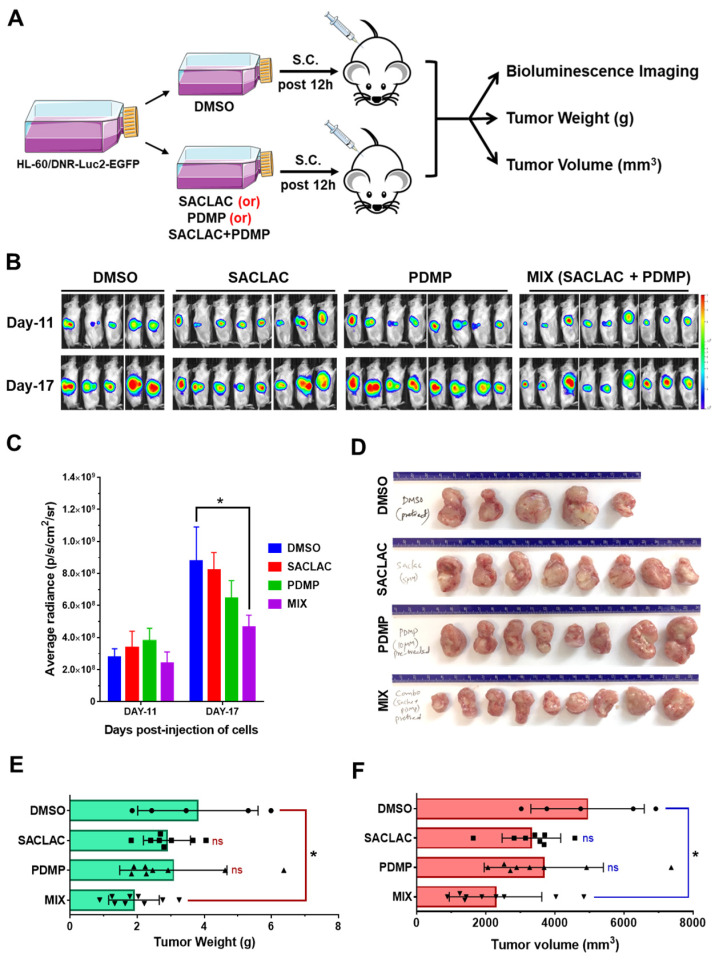
Exposure of HL-60/DNR cells to combination SACLAC D-*threo*-PDMP ameliorated tumor progression in vivo. (**A**) HL-60/DNR-Luc2-EGFP cells were treated in vitro with either SACLAC (5 µM), D-*threo*-PDMP (10 µM), or the mix for 12 h; controls were treated with vehicle (DMSO). An equal number of viable cells (2.5 × 10^6^) from control and treatment groups (cell viability did not differ after 12 h pretreatment) were injected subcutaneously (s.c.) into 7–9 wk old male NRG-S mice (5–9 animals/group), and tumor progression was monitored by bioluminescence imaging (BLI), measurement of tumor weight (gm), and tumor volume (mm^3^). (**B**) BLI Images of mice on indicated days post-injection of pretreated HL-60/DNR-Luc2-EGFP cells injected s.c. (**C**) Quantification of BLI signals as average radiance (p/s/cm^2^/sr) for different treatment groups on indicated days. * *p* < 0.05 denotes statistical significance compared to the DMSO group by two-way ANOVA (Dunnett’s multiple comparisons test). (**D**), The tumors were harvested from different treatment groups for gross examination of tumor sizes. (**E**,**F**) Average tumor weight and tumor volume in different experimental groups were compared for statistical differences by one-way ANOVA (Tukey’s multiple comparisons test) * *p* < 0.05 was considered significant compared to the vehicle DMSO group.

**Figure 7 cancers-15-01883-f007:**
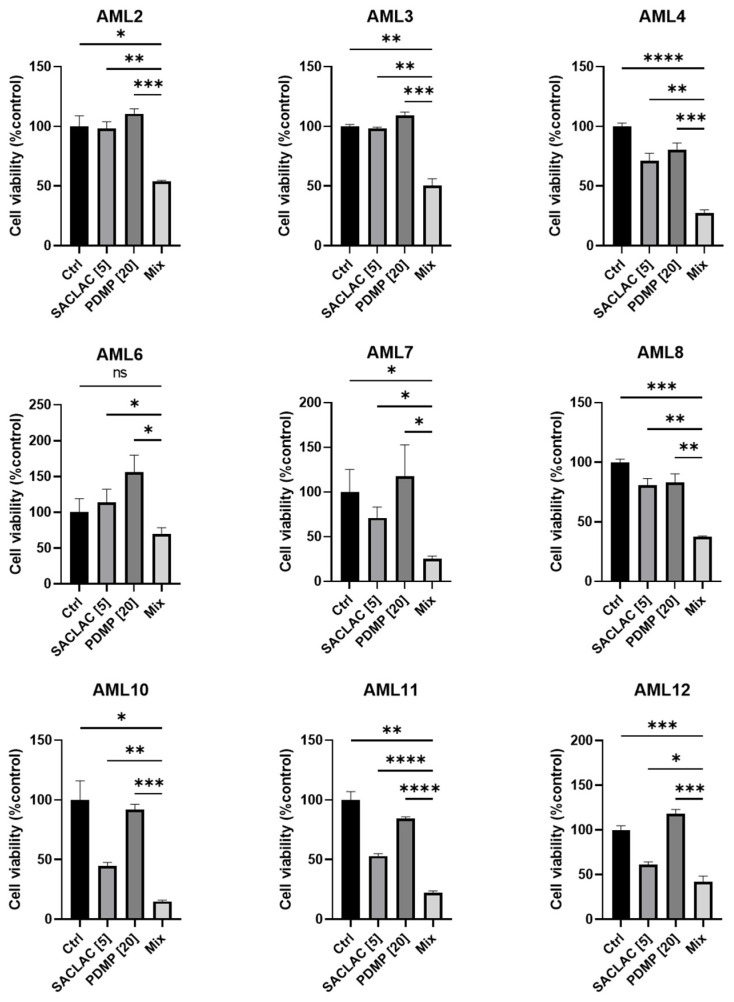
Combination SACLAC D-*threo*-PDMP treatment effectively reduces viability in primary de novo AML patient samples. Cell viability in a cohort of nine primary, previously untreated, de novo, AML patient PBMC samples treated with control (DMSO), SACLAC (5 μM), and/or PDMP (20 μM) for 48 h. Cell viability was measured using the CellTiter Glo luminescence assay. Luminescence was normalized to DMSO control treatment, which was set to 100%. Error bars represent the SD of three technical replicates. An unpaired *t*-test with Welch’s correction was used for statistical analysis (*, *p* < 0.05; **, *p* < 0.01; ***, *p* < 0.0005, ****, *p* < 0.0001). ns, not significant.

## Data Availability

Data are contained within the article and within the Appendix A.

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
