# Peer review of "Simultaneous Inhibition of Ceramide Hydrolysis and Glycosylation Synergizes to Corrupt Mitochondrial Respiration and Signal Caspase Driven Cell Death in Drug-Resistant Acute Myeloid Leukemia"

_cancers, 2023, doi:10.3390/cancers15061883_

Round 1

Reviewer 1 Report

The paper is very well written with solid experimentation and good controls.  The experiment in Figure 1 is excellent, especially since you tested whether this is a stable resistant phenotype or not.  However if you could clarify what each of the the figures (left and right) on Figure 1 A and B are (I assumed it was the full dose response and then the right panel was jus 0 to 1.0 ug/ml expansion of the left panel), that clarity in figure legend would be beneficial to reader.

It might be useful in discussion to explain why combination does not fully kill off cells and what other survival mechanisms might be upregulated in the resistant cells.  Also please in figure legend in 2B describe mix (what ratio of SACLAC to D-threo-PDMP).

Although not a weakness in the paper nor do additional experiments need to be done, the limitation in the animal studies, as you state, is the limited bioavailability of these compounds, therefore necessitating the pretreatment before implantation.  It might then be useful, for future reference, to just use organoid or just soft agar assay until the bioavailability issue can get resolved (unless a GEM is developed).  But this is not an issue for this paper.

Overall the conclusions are sound and will eagerly await the full publication of this manuscript.

Author Response

The paper is very well written with solid experimentation and good controls.  The experiment in Figure 1 is excellent, especially since you tested whether this is a stable resistant phenotype or not.  However if you could clarify what each of the the figures (left and right) on Figure 1 A and B are (I assumed it was the full dose response and then the right panel was jus 0 to 1.0 ug/ml expansion of the left panel), that clarity in figure legend would be beneficial to reader.

Another reviewer was also confused by this representation. In order to clarify, we omitted the low dose range 1-1.0 ug/ml and are only showing the 0 -10 ug/ml dose range. The figure legend has been slightly modified.

It might be useful in discussion to explain why combination does not fully kill off cells and what other survival mechanisms might be upregulated in the resistant cells.  Also please in figure legend in 2B describe mix (what ratio of SACLAC to D-threo-PDMP).

We have related in the Results section that the combination perhaps does not kill all the cells (in Fig. 2) because we did not carry the concentrations high enough, and that other survival mechanisms may be functioning, such as DNA damage repair, drug inactivation, or epigenetic changes. We have defined mix in the figure legend as the SACLAC + PDMP combination and given the ratio as 1:2.

Although not a weakness in the paper nor do additional experiments need to be done, the limitation in the animal studies, as you state, is the limited bioavailability of these compounds, therefore necessitating the pretreatment before implantation.  It might then be useful, for future reference, to just use organoid or just soft agar assay until the bioavailability issue can get resolved (unless a GEM is developed).  But this is not an issue for this paper.

This is an excellent suggestion, as new SACLAC formulations are surely needed. We have recently brought on a collaborator to help with formulating SACLAC.

Reviewer 2 Report

In this paper, the authors inhibited ceramide hydrolysis and ceramide glycosylation and found that this produced higher levels of intracellular ceramide, damaged mitochondrial function, increased caspase activation, and caused cell death in drug-resistant AML models. The study identifies targets in sphingolipid metabolism that can be used to enhance vulnerability of leukemia cells in drug-resistant cancers, providing a potential new therapy for drug-resistant AML.

The manuscript is clearly written and the conclusion are well supported by a good amount of data including in vivo validations. I have some suggestions before formally recommending it for publication:

1. In Figure 5, the authors showed the trend for basal respiration. Would it be possible to assess other oxygen fluxes (e.g. ATP-linked and max capacity)? If not, could the authors suggest possible changes on this in the discussion section?

2. Recent research in AML and sphingolipids (esp. ceramides) seemed to identify some sort of connections between ceramides (and other sphingoid bases), cellular pH and the pathology of AML. Would the authors be able to discuss how their own findings may improve our understanding on this potential mechanistic axis, using the following literature? (DOIs 10.1182/blood-2019-123604, 10.1016/j.jbiosc.2009.07.007, 10.1016/j.copbio.2022.102739, 10.1021/la5003397).

3. The authors may consider including some statistical assessment for Figure 2A (e.g. final point or total curve t-test or something along those lines).

Author Response

 In Figure 5, the authors showed the trend for basal respiration. Would it be possible to assess other oxygen fluxes (e.g. ATP-linked and max capacity)? If not, could the authors suggest possible changes on this in the discussion section?

 In the revised Fig. 5, panel ‘D’ now depicts intact cell respiration in HL60/DNR cells exposed to either DMSO or SACLAC/PDMP. Respiration was assessed under basal conditions, as well as in response to oligomycin, FCCP and Rotenone/Antimycin. Like we published for HL60/VCR (PMID: 34888943), the presence of p-glycoprotein renders HL60/DNR cells resistant to oligomycin. After correcting for oxygen consumption remaining following rotenone/antimycin, both basal and maximal respiration were decreased in HL60/DNR cells exposed to SACLAC/PDMP.

 Recent research in AML and sphingolipids (esp. ceramides) seemed to identify some sort of connections between ceramides (and other sphingoid bases), cellular pH and the pathology of AML. Would the authors be able to discuss how their own findings may improve our understanding on this potential mechanistic axis, using the following literature? (DOIs 10.1182/blood-2019-123604, 10.1016/j.jbiosc.2009.07.007, 10.1016/j.copbio.2022.102739, 10.1021/la5003397).

I have discussed this issue/suggestion with co-authors and we concluded that the information might not be useful in strengthening the presentation. We hope this does not detract in any way.

The authors may consider including some statistical assessment for Figure 2A (e.g. final point or total curve t-test or something along those lines).

Although we have not included statistical evaluation on the differential impact of PDMP and SACLAC on these cell lines, we have stated in the Results section under Fig. 2 that qualitatively, these differences are clear. We did not run up the concentration of PDMP high enough to calculate EC50's. We hope the qualitative assessment is sufficient to make evident the differences.

Reviewer 3 Report

Ceramide has been shown to induce apoptosis and limit cancer cell proliferation by many studies. Reprogramming metabolism of ceramide is common for cancer cells to mute its apoptotic response such as upregulating enzymes that metabolize ceramide. Drug resistance is a great scientific and clinical challenge in cancer treatment. It has been proposed that disruption of ceramide metabolism is one of the drug resistance mechanisms in several cancer types. Here, Fisher-Wellman et al. explored a combination treatment of ceramide hydrolysis and glycosylation to test if synergistic effects could be achieved using drug-resistant acute myeloid leukemia as an in vitro and in vivo model. The authors demonstrate that the ceramide hydrolysis inhibitor SACLAC and glycosylation inhibitor PDMP imposed a synergetic effect on the cell viability of AML in drug-resistant cell types but not in wild-type cells. They then used various approaches to further investigate possible mechanisms through profiling ceramide-derived sphingolipids, examining expression levels of proteins related to apoptosis pathways, and evaluating mitochondrial function. Lastly, they used the in vivo mouse model and patient samples to strengthen the conclusion. Overall, the findings are both interesting and potentially important.

Major issues:

  1. The data from the mouse experiment is not reliable as the authors did not implant the same number of viable cells in mice after cells were pretreated with inhibitors for 12 hours.
  2. The approach that the authors used for cell respiration assay was not really in physiological conditions. Instead, the Seahorse Mito stress test is a much better choice for this assay.
  3. It is not clear to me how the cell viability was calculated specifically in each figure as the different approaches were used to measure cell viability. For example, trypan blue exclusion gives a cell number. MTS presents a fluorescence signal instead of a cell number. Did the authors have a standard curve to convert the fluorescence signal into cell numbers?

Minor issues:

  1. Figure 1 is a little bit confusing. Is it possible to use two segments on the x-axis to present the data in the same figure instead of using two separated figures?
  2. The combination treatment decreased cell viability except for Molm 14 (Figure 2B). The authors didn’t explain this in the text.
  3. The western blotting in Figure 2E was overexposed. Please choose a different image that was exposed for less time if possible.
  4. In Figure 5B, it is not appropriate to use the same loading control for both 2-hour treated samples and 20-hour treated samples. I guess their lysates were harvested at different time points. Different b-actin loading controls should be used.
  5. The author may want to double-check the statistical analysis in all the figures. To me, it seems that there is a significant difference in the groups of Glu/Oct and delta-GATP(-41, 49) in Figure 5E.

Author Response

Major issues:

  1. The data from the mouse experiment is not reliable as the authors did not implant the same number of viable cells in mice after cells were pretreated with inhibitors for 12 hours.                                                                  Section 2.10 and Fig. 6 legend state that 2,500,000 viable cells were injected after being treated with the agents.
  2. The approach that the authors used for cell respiration assay was not really in physiological conditions. Instead, the Seahorse Mito stress test is a much better choice for this assay. 
    In revised figure 5D, we assessed respiration in intact cells under basal conditions, as well as in response to oligomycin, FCCP titration, and rotenone/antimycin. This assay was run in bicarbonate free RPMI-1640. We observed lower basal and maximal respiration in HL60/DNR cells treated with 'Mix', which agrees with the results of our permeabilized cell assay in 5E.
  3. It is not clear to me how the cell viability was calculated specifically in each figure as the different approaches were used to measure cell viability. For example, trypan blue exclusion gives a cell number. MTS presents a fluorescence signal instead of a cell number. Did the authors have a standard curve to convert the fluorescence signal into cell numbers? We specifically used cell counting and trypan blue exclusion to determine viable and dead cell numbers. This method was used for final verification of the impact of the various treatment regimen. We agree that MTS works via fluorescence, never-the-less, this method is a good representation of viable cells, as dead cells will not generate color. We believe that these methods are reliable for determining percent viable populations. We did not make a standard curve to calculate cell number from fluorescence readings.

Minor issues:

  1. Figure 1 is a little bit confusing. Is it possible to use two segments on the x-axis to present the data in the same figure instead of using two separated figures? Another reviewer said the same, we therefore removed the panels showing 0-1.0 ug/ml and now only show 0 - 10 ug/ml.
  2. The combination treatment decreased cell viability except for Molm 14 (Figure 2B). The authors didn’t explain this in the text. We have added to the text: The unusual MOLM 14 response, stimulatory with the combination, is perhaps related to generation of mitogenic lipids such as sphingosine 1-phosphate.
  3. The western blotting in Figure 2E was overexposed. Please choose a different image that was exposed for less time if possible. We are sorry for this predicament; however, Western blots for P-gp are often like this because expression is gigantic in MDR cells. We decided not to lighten the background.
  4. In Figure 5B, it is not appropriate to use the same loading control for both 2-hour treated samples and 20-hour treated samples. I guess their lysates were harvested at different time points. Different b-actin loading controls should be used. We are sorry for this. The loading control for the 2 hr is shown in the Supplementary data.
  5. The author may want to double-check the statistical analysis in all the figures. To me, it seems that there is a significant difference in the groups of Glu/Oct and delta-GATP(-41, 49) in Figure 5E. We have corrected the stats in Fig. 5E. Thank you for raising this point.

Reviewer 4 Report

Fisher-Wellman et al reported a combined treatment with ceramide hydrolysis and glycosylation inhibitors on drug-resistant AML cells and found increased intracellular ceramides level, diminished mitochondrial respiration and enhanced apoptosis. This is attractive to the field of resistant AML, particularly novel treatment regimens.

Major points:

1. How are the concentration of SACLAC (5 µM) and PDMP (10 µM) determined? The current dosage shows evident synergism and profound cytotoxic effects on AML cells. However, this also induces 20%~30% cell death in PBMCs, which would be assumed from normal, healthy samples. This may be a major concern in translation into the clinical. Have the authors tried combination at lower dosage?

2. The authors suggest C16:0 ceramide is pro-apoptotic while longer ceramides C24:1 and C26:1 are anti-apoptotic, and C16:0/C24:1 ratio is proposed as pro-survival indicator. While other ceramides, including “longer” C18, C20 and C22 are also upregulated, could the author further elaborate whether the rise of their levels are driving factors or cell-saving response, as well as possible roles of different ceramide species? Is there a possible way to manipulate specific ceramide intracellularly?

3. Figure 4 also shows synergism of SACLAC and other P-gp antagonists. Then what could be the advantages of SACLAC + PDMP over SACLAC + P-gp antagonist? Apparently, the mechanism of latter is more defined.

4. In the mouse experiments, pre-treated cells were injected into mice and no further SACLAC or PDMP treatment was applied, as seen from Figure 6. This is conceptually different from a treatment regimen but more like a prophylaxis. Would the mice injected with pre-treated HL-60/DNR cells have better response to DNR? It would also be informative if the mice were injected with HL-60/DNR cells and then treated with SACLAC+PDMP with or without DNR.

Minor points:

1. For panels A&B in Figure 1, are line charts on the left and the right from the same set of experiments, while a different x-axis is used to show lower dose range? If so, it’s better to clarify this point in the figure legends. Or, alternately, a log-scale may be used for x-axis.

2. P-gp is suggested to be one major target of the combining treatment, and an investigation of p-gp expression is shown in Figure 1E, but a control from PBMC is missing. Is the presence of P-gp a feature of drug-resistant AML only?

3. The western blot image of Mcl-1s and cleaved Cas-2 (Figure 5B) seems different from original images in the supplemental figure, in terms of the ratio of width and distance between major/minor bands. Please double check it.

4. For DAPI/Annexin V results by flow, the prominent change can be seen in the late apoptosis quadrant. Has the authors repeated the experiment at an earlier stage to catch up any difference in early apoptosis?

5. Regarding mitochondrial respiration assays, the authors suggest that mitochondrial membrane integrity is compromised upon “mix” treatment. Could the authors provide any evidence of this point, like isolation of mitochondria and detection of outer membrane proteins? Could authors exclude the possibility that digitonin destroys outer membrane as a result of increased sensitivity to this detergent in mix treated HL-60/DNR cells?

Author Response

1. How are the concentration of SACLAC (5 µM) and PDMP (10 µM) determined? The current dosage shows evident synergism and profound cytotoxic effects on AML cells. However, this also induces 20%~30% cell death in PBMCs, which would be assumed from normal, healthy samples. This may be a major concern in translation into the clinical. Have the authors tried combination at lower dosage? The SACLAC and PDMP concentrations were determined by running preliminary kill curves (dose-response viability); from these we chose the 5 and 10 uM values. We agree there was some cytotoxicity in PBMC's at 5 + 10, but we do not have data on lower concentrations. This is a good point for future clinical application. 

  1. The authors suggest C16:0 ceramide is pro-apoptotic while longer ceramides C24:1 and C26:1 are anti-apoptotic, and C16:0/C24:1 ratio is proposed as pro-survival indicator. While other ceramides, including “longer” C18, C20 and C22 are also upregulated, could the author further elaborate whether the rise of their levels are driving factors or cell-saving response, as well as possible roles of different ceramide species? Is there a possible way to manipulate specific ceramide intracellularly? These are good points. In the MDR phenotype HL-60/DNR (compared to wild type HL-60), Fig. 3A, the rise in 24:1 may be a survival factor in DNR resistance. As for the increases in C18, C20 C22 ceramides in treated cells, Fig. 3B, we posit that this contributes to cell death. As there is ceramide molecular species overlap regarding pro- versus anti-apoptotic effects, and cell-type specificity, direct conclusions are difficult to draw. Ceramide synthase knockouts can be employed and perhaps supplementing the media with specific fatty acids (C24:1 for example) may shift the profiles of intracellular ceramide molecular species.
  2. Figure 4 also shows synergism of SACLAC and other P-gp antagonists. Then what could be the advantages of SACLAC + PDMP over SACLAC + P-gp antagonist? Apparently, the mechanism of latter is more defined. As related in the Conclusions, we believe halting ceramide glycosylation at P-gp is better to boost the "ceramide effect". We have cited several works that show GCS inhibitors like PDMP act at P-gp and can even block chemotherapy efflux in MDR cells. Repurposing P-gp antagonists in this case, is promising.
  3. In the mouse experiments, pre-treated cells were injected into mice and no further SACLAC or PDMP treatment was applied, as seen from Figure 6. This is conceptually different from a treatment regimen but more like a prophylaxis. Would the mice injected with pre-treated HL-60/DNR cells have better response to DNR? It would also be informative if the mice were injected with HL-60/DNR cells and then treated with SACLAC+PDMP with or without DNR. We are currently designing novel SACLAC formulations, as solubility and bioavailability make in vivo administration difficult. That is the reason we pre-treated the cells before injection. We have not determined whether the pre-treated injected HL-60/DNR cells would have a better DNR response. That would be interesting to determine. The different approaches suggested are good avenues that await our new SACLAC formulations.

Minor points:

  1. For panels A&B in Figure 1, are line charts on the left and the right from the same set of experiments, while a different x-axis is used to show lower dose range? If so, it’s better to clarify this point in the figure legends. Or, alternately, a log-scale may be used for x-axis. Three out of 4 reviewers had difficulty with this figure. We have modified it to present only 1 - 10 ug/ml dosage, omitting the 0 -1.0 ug panels.
  2. P-gp is suggested to be one major target of the combining treatment, and an investigation of p-gp expression is shown in Figure 1E, but a control from PBMC is missing. Is the presence of P-gp a feature of drug-resistant AML only? Upregulated P-gp is a major feature of drug resistance in all cancers, but usually in response to drugs such as anthracyclines, Vincas, taxanes, large heterocyclics. P-gp is found in normal colon but levels in PBMC's are extremely low and would not play a part in SACLAC PDMP cytotoxicity. It would be interesting to study ceramide generation in PBMC's in response to SACLAC and PDMP; we will plan this work.
  3. The western blot image of Mcl-1s and cleaved Cas-2 (Figure 5B) seems different from original images in the supplemental figure, in terms of the ratio of width and distance between major/minor bands. Please double check it. We checked this, and the image in Fig. 5B was squeezed to make boxes similar in height. We have corrected this in the revised figure, and not squeezed it.
  4. For DAPI/Annexin V results by flow, the prominent change can be seen in the late apoptosis quadrant. Has the authors repeated the experiment at an earlier stage to catch up any difference in early apoptosis? We have not looked at early apoptosis. The 24 hr time point and the 5 + 10 uM dose was used as this was the scheme in most of the work.
  5. Regarding mitochondrial respiration assays, the authors suggest that mitochondrial membrane integrity is compromised upon “mix” treatment. Could the authors provide any evidence of this point, like isolation of mitochondria and detection of outer membrane proteins? Could authors exclude the possibility that digitonin destroys outer membrane as a result of increased sensitivity to this detergent in mix treated HL-60/DNR cells? In the revised manuscript we included an outer-membrane integrity assay. In this assay, exogenous cytochrome C was added to digitonin permeabilized cells energized with saturating carbon substrates. An increase in respiration in response to cytochrome C is indicative of compromised outer-membrane integrity. In this assay, respiration was increased by cytochrome C only in HL60/DNR exposed to 'Mix', consistent with disrupted outer-membrane integrity. Although we cannot rule out that the effects were the result of differences in digitonin sensitivity, our results are 5E are supported by respiration data conducted in intact cells, now in 5D. 

Round 2

Reviewer 3 Report

The authors answers all my questions. Thanks.

Reviewer 4 Report

The authors have fully addressed my points and I'm happy to recommend acceptance of this manuscript.